# Gender and mental health of adolescents: A conceptual framework developed in a Delphi study

Anita Alaze[1]*, John Grosser[2], Oliver Razum[1], Céline Miani[1]

1 Department of Epidemiology and International Public Health, School of Public Health, Bielefeld University, Bielefeld, Germany, 2 Department of Health Economics, School of Public Health, Bielefeld University, Bielefeld, Germany

* anita.alaze@uni-bielefeld.de

## Abstract

### Objective

Adolescence is one of the most neglected developmental phases, yet there are few phases in the life course when gender socialisation is as intense. So far, no framework or theory exists that theoretically conceptualises gender norms, gender attitudes and mental health outcomes. To address this gap, we aim to develop a conceptual framework that illustrates the interplay of key constructs in the gendered pathways of health, using the Delphi technique.

### Methods

We carried out a Delphi study enrolling an international panel of 21 experts from different disciplines, with 43% of the respondents completing all three survey rounds. We asked the experts to identify core constructs (first round), suggest operationalisations of the constructs (second round), and to hypothesise about their relationships (third round). Items were included, excluded or adapted according to experts' feedback. A 70% threshold was used for consensus.

### Results

The panelists consented on the following core constructs: gender norms (social environment), sex assigned at birth, gender attitudes, gender roles, competencies, and gender identity to the mental health of adolescents. They further consented on gender approaches, namely the intersectional, multidimensional and multilevel approach as well as the power relations lens. The operationalisation of the constructs in round two led to the inclusion of variables forming an intersectional lens and four social environment levels: the household, community, political and digital level. In round three, the experts formed 14 hypotheses about the relationships between the core constructs.

**Data availability statement:** The Data Availability statement is under discussion and will be provided in a forthcoming update to this article.

**Funding:** This paper is part of a PhD and was thereby funded within the scope of the Junior Research Group GendEpi (Gender Epidemiology) within the Department of Epidemiology & International Public Health, School of Public Health, Bielefeld University. The second funding phase of the Junior Research Group was funded by Bielefeld University [grant number not applicable, to CM]. The funders had no role in study design, data collection and analysis, decision to publish, or preparation of the manuscript.

**Competing interests:** The authors have declared that no competing interests exist.

## Conclusion

The derived conceptual framework links the interplay of six key constructs, four gender approaches and four social environment levels to adolescent mental health. Future research should validate and apply the hypothesised relationships between the constructs to disentangle the gendered pathways to (mental) health in adolescence.

## Introduction

Gender norms, the "spoken and unspoken rules of societies about the acceptable behaviours of girls and boys […] - how they should act, look, and even think or feel" [1], contribute to sex differences in morbidity and mortality in adolescents. Gender norms can lead to gender inequalities which manifest in unequal access to resources, power, opportunities, education or discriminatory cultural practices [2], hierarchical social structures or harmful gender stereotypes [3]. Inequitable gender norms are increasingly seen as a risk to health [4]. They have implications for the *gendered pathways to health,* in which gender norms encourage or discourage specific health behaviours, foster unequal power dynamics, and create inequitable access to services, which can lead to adverse health outcomes [1,5]. Gender norms are acquired through various actors in the adolescents' social environment, such as the family, the school environment, including peers, and the media. In the following, we describe this process and the relation of gender norms with adolescents' mental health.

The gender socialisation process, a "process by which individuals learn the cultural behaviours associated with the concepts of femininity or masculinity" [6], begins primarily in the family where parents serve as crucial agents of socialisation [7]. Parents of every cultural setting directly or indirectly influence their children to conform to expected gender roles. The resulting gender-typed behaviours in children can occur even before the age of two years [8]. These gender expectations are rooted in gender norms [1].

Besides the family, the *school environment* can influence the gender socialisation process of children and adolescents. The school environment can reinforce gender norms established at home, such as through the use of books that depict inequitable gender norms or teachers assigning gendered roles to students [7]. Peer group gender norms are incorporated into the self-concept, "a structured, multidimensional, and evolving construct that constitutes all the beliefs that an individual has about him/herself" [9]. This makes the peer group central to the construction of gender attitudes [2].

Additionally, the use of *(social) media* can lead to a disparity between the adolescents' perceived reality and the content shared on social media. This disparity can be exacerbated by misuse or intensive use of social media [10].This can lead to body dissatisfaction, low self-esteem, risky behaviours, and eating disorders [11].

While the gender socialisation process lasts throughout the life course, it becomes more pronounced during critical periods of development, such as early adolescence (10–14 years). In this period, social, cognitive, and physical functions develop rapidly

[5,12]. Adolescence is also a period in which rates of common mental health problems, such as depression or anxiety, rise substantially [13]. Recently, mental health in adolescents (10–19 years) has become a research priority due to an increase in the prevalence of mental health issues for this age group [14,15]. The situation is further exacerbated by the fact that many of these adolescents are not receiving the treatment they require [16]. Not only girls and gender minorities benefit from reducing inequitable gender norms, but also boys, who are likewise experiencing adverse health outcomes through harmful masculinity norms [17]. Despite the fact that adolescence is one of the few stages during the life course where changes are as dramatic and thus play a crucial role in understanding the gendered pathways to health, this period has been overlooked in research [12,18].

Previous research has predominantly focused on sexual and reproductive health, violence and HIV [1], leaving gaps in the links between gender norms, gender attitudes and various (mental) health outcomes. To address this gap, our research question for this study is as follows: What are key constructs in the gendered pathways of health, from gender norms to adolescent's mental health, and how do they interplay? Therefore, we aim to develop a conceptual framework that illustrates the interplay of key constructs in the gendered pathways of health, from gender norms to adolescent's mental health. Key constructs are mental health outcomes, competencies of adolescents in order to deal with (internalised and externalised) expectations of gender norms, as well as a range of gender concepts and approaches. In the following, we understand gender concepts as an umbrella term for gender norms, gender attitudes, gender roles, sex assigned at birth, and gender identity. We include the intersectionality approach, the gender relations lens, the multilevel approach, and the multidimensionality approach to the umbrella term of gender approaches. We distinguish between individual gender norms which we refer to as gender attitudes and the gender norms of the social environment, gender norms. To develop this framework, we employed the Delphi technique. The resulting conceptual framework is expected to provide a basis for more theory-driven research in this field.

## Materials and methods

### Delphi study design

The Delphi technique is a systematic and iterative expert-based assessment process carried out by a group of experts to find consensus for an epistemic question [19]. This qualitative technique has been widely used in public health and mental health research [20–22] and has also been applied in social epidemiology (e.g., [23]). Scholars have utilized the Delphi approach to develop frameworks and achieve consensus on various health agendas [24,25], particularly for topics with scarce evidence [26]. A key feature of the Delphi technique is that after each round, the experts receive the results of the previous round so that they can potentially change their assessments [19].

In this study, we employed the Delphi technique with a continuous panel over three online Delphi rounds and a threshold of 70% agreement to gain consensus on a conceptual framework. An online Delphi survey allows the experts to think longer about certain aspects, which could contribute to more thoughtful and balanced contributions. The online survey is also advantageous for a panel with experts from various geographical locations [27]. Another important aspect of this Delphi is the anonymity of the panelists. This removes biases such as dominance or group conformity and ensures independent objectivity [28], which is particularly important in an interdisciplinary research field, such as ours.

Our study adheres to CREDES recommendations (Guidance on Conducting and REporting DElphi Studies) [28], which were developed for palliative care research but apply to the Delphi technique in general (see Supporting Information S1 Table). Fig 1 provides the schematic for our Delphi study design.

Before the first Delphi round, we reviewed the literature of relevant frameworks and set up a pool of questionnaire items (explained in more detail in the following subchapter) which we then revised in an iterative process. The term item refers to a questionnaire question along with its corresponding answer options. The iterative process included conducting cognitive interviews with experts to assess the relevance and completeness of the content in the questionnaire, assessing the suitability of the first questionnaire by Delphi experts and holding group discussions with experts from different disciplines

| Development process of the Delphi study design | | | |
|---|---|---|---|
| ➢ Conducting Delphi round 1<br>➢ Analysis of Delphi round 1<br>➢ Development of questionnaire 2 based on results of round 1<br>➢ Provision of feedback document to Delphi panel | ➢ Conducting Delphi round 2<br>➢ Analysis of Delphi round 2<br>➢ Development of questionnaire 3 based on results of round 2<br>➢ Provision of feedback document to Delphi panel | ➢ Development of questionnaire 3 based on results of round 2<br>➢ Conducting Delphi round 3<br>➢ Analysis of Delphi round 3 | ➢ Interpretations and conclusions of Delphi rounds<br>➢ Development of final conceptual framework<br>➢ Provision of feedback document to Delphi panel |

**Fig 1. Development process of the Delphi study design.**

to further revise and refine the questionnaire. We have described the conduct and results of the development process of the first Delphi round extensively in a previous publication [29].

We recruited experts via email and conducted three Delphi rounds. After each Delphi round, we analysed the quantitative and qualitative results. Based on the results of the previous round and the goal of developing a conceptual framework within three Delphi rounds, we created the questionnaire for the following rounds. We granted the panelists two weeks to complete the survey and extended it for another week if we received no response. Before each due date, we sent two reminders. If we still did not receive a response in rounds two and three, we asked the experts to indicate a date when they would be able to complete the survey, as they had agreed to participate in the following rounds. This was necessary to ensure a minimum number of approximately ten participants in the survey and thus obtain valid results. Hence, each round took about eight weeks to complete. After each round, we provided feedback about the findings to the panelists.

After the three Delphi rounds, we drew conclusions, summarised the results of the three rounds and developed the final conceptual framework, which we ultimately reported back to the participants. The research team planned and managed the overall study process and analysed and interpreted the results of each round in order to incorporate them into the next Delphi round. Complex and difficult decisions on the design and outlay of the following Delphi round were discussed and revised between rounds. The research team carefully reviewed and piloted all material provided to the expert panel throughout the whole Delphi process.

## Content of Delphi rounds

Each of the three Delphi questionnaires consists of three substantive sections: A section on gender, a section on mental health and a section on the social environment. In the first questionnaire, a fourth section on the socio-demographic characteristics of the participants has been added. Table 1 presents an overview of the content covered in each Delphi round, along with the data collection technique employed in each round. The supporting information shows the second and third questionnaires in more detail (Supporting Information S4, S5 Table).

In the *first Delphi round*, we aimed at developing a conceptual framework that includes a) mental health as an outcome, b) sex assigned at birth and gender identity as two essential elements for a sex/gender constellation that is not binary, and (c) a set of other gender-related constructs (gender concepts and approaches). Thus, we introduced key gender concepts, such as gender roles, gender attitudes, gender norms and gender identity. Further, we asked participants to rate various gender approaches by their level of importance, such as the multilevel approach or the intersectionality lens. We also asked the experts to provide gender norms that particularly influence adolescents' mental health. In the mental health section, the experts were required to rate the proposed mental health outcomes by their level of importance to capture the impact of gender norms on adolescent mental health. In the social environment section, we proposed five different social environment levels in an illustration and asked whether these levels are the most relevant for the age group of adolescents. In addition, we proposed actors or stakeholders that may play a role in the gender socialisation process and/ or the mental health of adolescents. Thereafter, these actors should be assigned to the social environment levels in which

**Table 1. Overview of the content of the three Delphi rounds and data collection techniques.**

| | Delphi round 1 | Delphi round 2 | Delphi round 3 |
|---|---|---|---|
| Aim of Delphi round | Selecting constructs for the framework | Proposing operationalisations for the constructs | Hypothesising (causal) relationships between the constructs |
| Gender section | Selecting key gender concepts and gender approaches; Listing gender norms | Providing a (validated) instrument or suggesting possible questionnaire items that cover(s) the consented construct; suggesting how the consented gender approaches can be reflected in the framework; Selecting gender norm conceptualisations | Rating adequacy of social position variables; identifying (causal) assumptions for the relationship between the constructs |
| Mental health section | Rating mental health outcomes | Providing a (validated) instrument or suggesting possible questionnaire items that cover(s) the consented construct | Identifying (causal) assumptions for the relationship between the constructs |
| Social environment section | Rating social environment levels; assigning actors/stakeholders to social environment levels; rating competencies of adolescents | Rating updated social environment levels | Identifying (causal) assumptions for the relationship between the constructs; assigning actors/stakeholders to the revised social environment levels |
| Socio-demographic characteristics | Year of birth, sex assigned at birth, current gender identity, sexual orientation, country with the most working experience, current country, professional environment, level of confidence in the content areas, introducing pseudonym | – | – |
| Data collection technique according to [30] | Likert scale ranking; qualitative/quantitative combination; listing ideas; rank preferences; short answers to open-ended questions; comment on pre-defined questions | Agree or disagree statements; multiple choice; qualitative/quantitative combination; selection; Short answers to open-ended questions; comment on pre-defined questions | Agree or disagree statements; multiple choice; qualitative/quantitative combination; selection; short answers to open-ended questions; comment on pre-defined questions |

they play the greatest role. Furthermore, we suggested various competencies that may be necessary for adolescents to cope with the gender norms of their social environments, and which should be rated according to their importance. For each construct, we allowed the experts to provide comments on their ratings by open text. After each section, we asked the participants if there was anything they would like to add that had not yet been addressed. We also gathered socio-demographic data (e.g., year of birth, sex assigned at birth, current sex/gender identity, sexual orientation, the country with the most working experience, current country, professional environment, and level of confidence in the content areas) and introduced a pseudonym for each panelist.

In the *second Delphi round*, we provided an overview of the constructs that had achieved consensus in the first Delphi round. We also reintroduced the constructs that had not reached consensus in the initial round and asked the experts whether, and which of these should be included in the conceptual framework. Moreover, we aimed at operationalising the previously selected constructs. Hence, for each selected gender concept, mental health outcome and competency, we asked the experts to either provide a (validated) instrument or to suggest possible questionnaire items that cover(s) the selected construct. These filter questions were included to reduce the time burden on participants. Another question was how the selected gender approaches (e.g., the multilevel approach) can be reflected in the conceptual framework. These approaches are not presented as their own constructs in the conceptual framework. Instead, they are reflected by other constructs (gender, gender norms, social environment levels, mental health) and the way these constructs are represented in the framework, i.e., the multilevel approach is reflected by the different social environment levels. For gender norms, we proposed two suggestions on categorisations of gender norms based on the results in round one and let the experts choose which suggestion they prefer (see Supporting Information S4 Table). In the social environment section, we

showed an updated illustration of the social environment levels and requested, whether the participants agreed to include it into the framework. As in the first questionnaire, we also provided comment fields and space for the experts to mention aspects that were not addressed.

In the *third Delphi round*, we presented the participants with an initial version of the conceptual framework. Moreover, social position variables were presented forming an intersectional perspective. Second, we gave an overview of the included aspects and how they could be implemented in the final framework. We allowed the experts to comment on these statements. In this round, our objective was to form hypotheses about the relationships between the constructs. The experts were thus asked to hypothesise (causal) relationship between the constructs, e.g., "sex assigned at birth has an influence on gender roles". Further questions concerned the operationalisation of gender roles and the adequacy of the social position variables in terms of relevance to adolescents and completeness. Since the names of the social environment levels reached consensus in the second round, we repeated the question we had introduced in the first round, namely which actors (as carriers of gender norms) are influencing adolescents' gender attitudes and to which social environment level they belong. We included further suggestions for actors from the panelists in the first round.

## Expert inclusion criteria, identification, and recruitment

Our purposeful sampling strategy targeted expert knowledge. The predefined criteria for this Delphi panel were: (1) researcher or working in development, services, or implementation (e.g. NGOs, policy, services and implementation, civil society organisations, governmental agencies, intergovernmental organisations), (2) sufficient written English and computer skills to contribute relevant input and communicate ideas effectively via an online questionnaire, (3) working (or having worked) on at least one of the following topics: gender, gender norms, gender socialisation, adolescent health, mental health or intersectionality, and (4) willingness and availability to complete at least three Delphi rounds. We also distributed our study in public health networks or asked people for nominations that fit our inclusion criteria.

We paid particular attention to geographical diversity, ensuring that experts from the Global South and women are not underrepresented, that experts of different ages are included and that, where possible, gender non-binary experts are included. This was done to ensure that the conceptual framework has incorporated diverse perspectives on the research topic since its development.

We recruited the experts with an email to the identified experts or the networks including an invitation, explanation of and link to the study. We contacted experts via LinkedIn when we could not find an email address. For the Delphi sample, we conducted several waves of recruitment before the first round until at least 20 experts participated in the study. Participants were not renumerated for their participation in this Delphi study. We have offered the participants the opportunity to use the conceptual framework once it has been developed.

## Definition of consensus

A 70% threshold was established for consensus. This required that at least 70% of the respondents indicated that the item is at least fairly important. For the different types of answer options, this means that the item is either 'fairly important' and/or 'very important' when we asked about the level of importance, 'most important' and/or 'second most important' when we asked ranking questions or at least 70% agreement to include or exclude an item.

The 70% threshold was not entirely defined a priori because of the explorative nature of the study. However, we predefined that the threshold should at least be higher than 50%. We have decided on the 70% threshold for consensus after the first round and then applied it to all three Delphi rounds. The 70% threshold is not extremely rigorous, nor is it a mere reflection of a simple majority, which we found suitable for our explorative and interdisciplinary research topic.

When no consensus could be reached for items in the first round, they were reintroduced in the second round as a list of remaining items. Among these, we asked the respondents to indicate which further items should be included in the

framework. The items on which no consensus could be reached after the second round were excluded. After the third round, we only included those aspects in the final conceptual framework that had reached a consensus.

### Feedback provision to inform expert's judgement

We fed back the quantitative results to the experts in the form of bar charts, where they could see their own answers compared to the other experts. We presented the qualitative results in the form of bullet points. This way, they were able to compare their score with the overall results and tendencies of the Delphi panel, in order to help achieve consensus in the subsequent Delphi round. In the feedback form of the first Delphi round, we also included the main (aggregated) socio-demographic characteristics of the Delphi panel.

### Data analysis

The Delphi study included both quantitative and qualitative data. We analysed the quantitative data descriptively using the statistical software R version 4.3.2. We calculated means, medians, standard deviations, and level of agreement (as percentages) and visualised the distribution of frequencies in bar plots for each questionnaire item.

Qualitative findings from the open-ended questions were used to inform the development of questionnaire items in the subsequent Delphi round, such as rewording items or introducing new items. All data were analysed similarly throughout the Delphi rounds.

### Ethical considerations

Ethical approval was obtained from the ethics board at Bielefeld University, Germany, in May 2023 (application number: 2023−111 of 2023/05/22). We informed the experts about the aim, their scope of participation, the number of rounds, procedures, data privacy, anonymity as well as potential risks and benefits of the study. We explained that the study was voluntary and that they could withdraw at any time without consequences. We obtained informed consent in written form via the online questionnaire. The panelists had to agree to the conditions in the first section of the questionnaire before they could proceed with the survey. This means that only those who accepted the conditions completed the survey. The recruitment period, including all three Delphi rounds, started on October 9, 2023 and ended on September 17, 2024.

## Results

### Delphi sample

The median age of the initial 21 Delphi experts was 39 years, and the majority of the panel was female (67%), cis-gender (71%) and heterosexual (57%; see Table 2). Two experts identified as gender non-binary and queer. Nine (out of 21) experts identified as having a sexual orientation other than heterosexual, namely lesbian or gay, bisexual, or preferred not to say. Slightly more than half of the experts had non-research-related backgrounds (52%). "Other" professional environments included clinical practice (psychiatry) and education/teaching (tertiary level). The regions in which respondents gained most of their working experience were Europe, Africa, the Asia-Pacific, and North America. This results in 38% of the participants having extensive working experience in countries that could be classified as the Global South. However, there was a tendency towards the inclusion of experts from the Global North and in particular German experts (29%). The self-rated levels of competence of the experts were intermediate in all areas, ranging on average between 'somewhat confident' and 'fairly confident' with the highest self-rated competence in gender overall, followed by mental health and gender norms.

To recruit experts for the study, we contacted 120 experts directly via email, and 14 experts via LinkedIn. We also distributed the study in 6 networks. Out of a total of 134 directly contacted experts, 21 experts completed the first Delphi round. This is a response rate of 16%, if we neglect the unknown number of eligible experts contacted through the

**Table 2. Descriptive statistics of the Delphi sample (n = 21).**

| | | Sample distribution | | | |
|---|---|---|---|---|---|
| **Country working experience** | **n** | **valid %** | **Age (median)** | **mean (SD)** | **(range)** |
| Austria | 1 | 5 | | 39 (12.03) | (30-69) |
| Germany | 6 | 29 | **Level of competence** | **mean (SD)** | **(range)** |
| United Kingdom | 3 | 14 | Gender | 3.86 (0.96) | (1 –5 ) |
| United States of America | 3 | 14 | Gender theory | 3.33 (1.24) | (1 –5) |
| Cameroon | 1 | 5 | Gender norms | 3.48 (1.03) | (1 –5) |
| Kenya | 1 | 5 | Gender socialisation | 3.05 (1.16) | (1 –5) |
| Zambia | 1 | 5 | Intersectionality | 3.52 (1.12) | (1 –5) |
| Ghana | 1 | 5 | Mental health | 3.62 (1.2) | (1 –5) |
| Egypt | 1 | 5 | Mental health of adolescents | 3.38 (1.36) | (1 –5) |
| Nepal | 1 | 5 | **Sexual orientation** | **n** | **valid %** |
| Indonesia | 2 | 10 | Heterosexual | 12 | 57 |
| **Current country** | **n** | **valid %** | Lesbian or Gay | 3 | 14 |
| Austria | 1 | 5 | Bisexual | 4 | 19 |
| Germany | 8 | 38 | Preferred not to say | 2 | 10 |
| Netherlands | 1 | 5 | Or please specify | 0 | 0 |
| United Kingdom | 1 | 5 | **Professional environment** | **n** | **valid %** |
| United States of America | 3 | 14 | Research | 10 | 48 |
| Canada | 2 | 10 | Policy | 6 | 29 |
| Cameroon | 1 | 5 | Development, services and implementation | 2 | 10 |
| Ghana | 1 | 5 | Other | 3 | 14 |
| Indonesia | 2 | 10 | **Current gender identity** | **n** | **valid %** |
| Nepal | 1 | 5 | Female | 14 | 67 |
| **Sex assigned at birth** | **n** | **valid %** | Male | 5 | 24 |
| Male | 6 | 29 | Non-binary | 1 | 5 |
| Female | 15 | 71 | Queer | 1 | 5 |
| Intersex | 0 | 0 | Trans*/transman/transwoman | 0 | 0 |
| | | | Inter* | 0 | 0 |
| | | | An identity not mentioned here | 0 | 0 |
| | | | I do not want to classify as any sex/gender category | 0 | 0 |

networks. We could not recontact one person who had participated in the first round again in the subsequent rounds, because of a missing email address. Out of the 20 experts recontacted after round 1, 12 completed Delphi round two, which led to a response rate of 60%. Delphi round three was completed by nine experts. The completion rate of the whole Delphi study (defined as the number of respondents filling in the first survey divided by the number of respondents filling in all three surveys) was 43%.

## Delphi procedure results: round 1

Table 3 shows the consensus percentage for each item that was quantitatively measured in round 1. In the section on gender in the first round, no gender concept reached consensus. However, we defined gender identity and sex assigned at birth as essential before the first round, which is why they were included in the framework despite not reaching consensus. As for the gender approaches, multidimensionality, multilevel, intersectionality, as well as the power relations lens reached a consensus. Six mental health outcomes were included, namely mental, social and physical well-being,

**Table 3. Consensus percentage for the quantitative results of the first Delphi round (n = 21).**

| Section | Questionnaire items | Consensus % (n) |
|---|---|---|
| Section A: Gender | | |
| Gender concepts | Importance of current sex/gender identity | 55 (12) |
| | Importance of sex/gender roles | 40 (8) |
| | Importance of sexuality | 40 (8) |
| | Importance of sex/gender expression | 37 (7) |
| | Importance of sex/gender relations | 21 (4) |
| Gender approaches | Importance of gender continuum | 52(11) |
| | Importance of multidimensionality approach | **90 (19)** |
| | Importance of multilevel approach | **95 (20)** |
| | Importance of intersectionality approach | **90 (19)** |
| | Importance of gender power relations lens | **76 (16)** |
| | Importance of embodiment approach | 48 (10) |
| | Importance of decolonial lens | 38 (8) |
| Section B: Mental health | | |
| Mental health outcomes | Importance of mental, social and physical well-being | **90 (19)** |
| | Importance of depressiveness | **76 (16)** |
| | Importance of connectedness | **71 (15)** |
| | Importance of body image | **86 (18)** |
| | Importance of self-efficacy | 43 (9) |
| | Importance of self-control | 43 (9) |
| | Importance of sense of coherence | 43 (9) |
| | Importance of happiness | **71 (15)** |
| | Importance of life purpose | 48 (10) |
| | Importance of suicidal behaviour | 57 (12) |
| | Importance of resilience | 52 (11) |
| | Importance of risky behaviour | **86 (18)** |
| | Importance of substance misuse | 67 (14) |
| Section C: Social environment | | |
| Social environment levels | Other social environment levels than the proposed 5 (see [a])? | Yes: 52 (11) |
| Inclusion of actors most likely to influence gender attitudes | Family – interrelational level | 76 (16) |
| | Peers – interrelational level | 81 (17) |
| | School environment- community level | 48 (10) |
| | Sport club – Community or interrelational level | each: 48 (10) |
| | Faith club – community level | 67 (14) |
| | Media – national level | 29 (6) |
| | Social media – global level | 33 (7) |
| | Healthcare providers – community level | 57 (12) |
| | Law enforcement – national level | 62 (13) |
| | Political parties – national level | 81 (17) |
| | Civil Society Organisations – national level | 38 (8) |
| | Non-Profit Organisations – national level | 43 (9) |

*(Continued)*

**Table 3.** (Continued)

| Section | Questionnaire items | Consensus % (n) |
|---|---|---|
| Adolescents' competencies | Importance of coping skills | **90 (19)** |
| | Importance of agency | **81 (17)** |
| | Importance of navigation | 52 (11) |
| | Importance of interpersonal relationship skills | **86 (18)** |
| | Importance of critical reflection skills | **90 (19)** |
| | Importance of mental health literacy | **76 (16)** |
| | Importance of respect and empathy | **76 (16)** |

Bold text = above defined consensus of ≥70% (fairly and very important)

depressiveness, connectedness, body image, happiness and risky behaviour. More than half of the participants (52.4%) were not satisfied with the proposed social environment levels. The actors as carriers of gender norms reached a consensus for three social environment levels: peers and family are suitable for the interrelational level and political parties for the national level. The participants found all but one of the proposed competencies, namely navigation, relevant to the framework.

The qualitative responses revealed that some gender concepts were unclear. The definition of the term sexuality, for instance, corresponded exactly to that of sexual orientation, which is why we changed the term to sexual orientation in the second round. Responses about the gender approaches showed that there is criticism of the gender continuum in the sense that it is not theoretically correct. Thus, we included the item 'gender spectrum' in the next round. The two questions on gender attitudes provided a rich and diverse cultural basis of specific attitudes that may have a relation with mental health (see Supporting Information S2 Table). The reactions concerning the proposed visualisation of the social environment levels led to changes in the names of the levels. Instead of *interrelational* we proposed *household,* and instead of *national and global* we proposed *political and digital* in the second round. Due to the renaming of the social environment levels, we repeated the question in which experts assign actors to these levels in the third Delphi round.

### Delphi procedure results: round 2

All constructs that did not reach consensus (≥70%) in round one, except for the two essential items (sex and gender identity), were reintroduced in the *second Delphi round* (see Table 4). This was done because a non-consensus does not directly imply a consensus on the exclusion of an item. In this round, two additional items reached a consensus: gender roles (83%) and resilience (75%). We also reached a consensus for the adapted social environment levels (75%), which are: individual, household, community, political and digital. The three participants who did not agree with the proposed levels did not provide an alternative suggestion. None of the newly introduced items based on the participants' comments in the first round achieved a consensus. Concerning the gender attitudes categorisations, the second suggestion reached a consensus (75%). Thus, we categorised gender attitudes into behaviour, body & appearance, sexual & relationship, performance, education, mobility and career attitudes and differentiate between descriptive norms (refer to what people of a certain gender are or what they do) and prescriptive norms (refer to what people of a certain gender *should* do).

The qualitative responses gave insights into how the participants would incorporate the gender approaches (multidimensionality, multilevel and intersectionality approach as well as the gender power relations lens) in the framework. Thus, we incorporate the multilevel approach by integrating several social environment levels. We include the multidimensionality approach by developing sex and gender concepts with different dimensions (sex assigned at birth, gender identity, gender roles). Lastly, we include both the intersectionality approach and the gender power relations lens by integrating multiple social positions forming an intersectional perspective for the population group of adolescents. These social

**Table 4. Consensus percentage of the quantitative questionnaire items of Delphi round 2 (n = 12).**

| Section | Questionnaire items | Consensus % (n) |
|---|---|---|
| Section A: Gender | | |
| Gender concepts | Importance of current sex/gender identity | N/A |
| | Importance of sex/gender roles | **83 (10)** |
| | Importance of sexuality (1st round)/ sexual orientation (2nd round) | 42 (5) |
| | Importance of sex/gender expression | 50 (6) |
| | Importance of sex/gender relations | 58 (7) |
| Gender approaches | Importance of gender continuum | 42 (5) |
| | Importance of gender spectrum | 33 (4) |
| | Importance of multidimensionality approach | N/A |
| | Importance of multilevel approach | N/A |
| | Importance of intersectionality approach | N/A |
| | Importance of gender power relations lens | N/A |
| | Importance of embodiment approach | 8 (1) |
| | Importance of decolonial lens | 50 (6) |
| Gender norms | Gender norms categorisations (Supporting Information S2 Table) | Suggestion 1: 17 (2)<br>**Suggestion 2: 75 (9)**<br>No suggestion: 8 (1) |
| Section B: Mental health | | |
| Mental health outcomes | Importance of mental, social and physical well-being | N/A |
| | Importance of depressiveness | N/A |
| | Importance of connectedness | N/A |
| | Importance of body image | N/A |
| | Importance of self-efficacy | 25 (3) |
| | Importance of self-control | 17 (2) |
| | Importance of sense of coherence | 25 (3) |
| | Importance of happiness | N/A |
| | Importance of life purpose | 33 (4) |
| | Importance of suicidal behaviour | 25 (3) |
| | Importance of resilience | **75 (9)** |
| | Importance of risky behaviour | N/A |
| | Importance of substance misuse | 33 (4) |
| Section C: Social environment | | |
| Social environment levels | Appropriateness (yes/no) of the proposed social environment levels? (see b) | **Yes: 75 (9)** |
| Adolescents' competencies | Importance of coping skills | N/A |
| | Importance of agency | N/A |
| | Importance of navigation | 17 (2) |
| | Importance of interpersonal relationship skills | N/A |
| | Importance of critical reflection skills | N/A |
| | Importance of mental health literacy | N/A |
| | Importance of respect and empathy | N/A |
| | Further inclusion of media literacy | 50 (6) |
| | Further inclusion of self-awareness | 50 (6) |
| | Further inclusion of life literacy | 0 (0) |
| | Further inclusion of self-efficacy | 25 (3) |
| | Further inclusion of spirituality | 8 (1) |
| | Further inclusion of assertive skills | 25 (3) |

[a]here, we proposed 5 different social environment levels: (individual), interrelational, community, national, global and the option not relevant. We put the level with the highest frequencies in the table.

[b]here, we proposed the levels: individual, household, community, political, and digital.

Bold text = above defined consensus of ≥70% (fairly and very important)

N/A = the question was not asked because the item reached consensus in the previous round

position variables encompass power dynamics and discrimination processes, addressing both individual-level factors and broader systemic gender inequalities at the macro level [31]. No panelist commented on the evaluation question of the questionnaire, but two participants reached out via email, indicating the need for a visualisation of the conceptual framework as a support to answer the questions.

### Delphi procedure results: round 3

In the *third round*, we reached a consensus for a number of hypothesised (causal) relationships (see Table 5), which are presented as arrows in the consented final conceptual framework in Fig 2. The operationalisation of gender roles, namely the time spent on gendered activities, achieved consensus. The items on social positions were also classified as relevant

**Table 5. Consensus percentage of the quantitative questionnaire items of Delphi round 3 (n = 9). This table displays only the consented items in round 3 (see table in full length in Supporting Information S3 Table).**

| Section | Questionnaire items | Consensus % (n) |
|---|---|---|
| Section A: Gender | | |
| Hypothesised causal influences starting from sex assigned at birth | Sex assigned at birth has an influence on gender attitudes | **Yes: 100 (9)** |
| | Sex assigned at birth has an influence on mental health | **Yes: 89 (8)** |
| | Sex assigned at birth has an influence on gender identity | **Yes: 78 (7)** |
| Hypothesised causal influences starting from gender identity | Gender identity has an influence on gender attitudes | **Yes: 78 (7)** |
| | Gender identity has an influence on gender roles | **Yes: 89 (8)** |
| | Gender identity has an influence on competencies | **Yes: 78 (7)** |
| | Gender identity has an influence on mental health | **Yes: 89 (8)** |
| Hypothesised causal influences starting from gender attitudes | Gender attitudes have an influence on gender roles | **Yes: 78 (7)** |
| | Gender attitudes have an influence on competencies | **Yes: 78 (7)** |
| | Gender attitudes have an influence on mental health | **Yes: 89 (8)** |
| | Gender roles have an influence on competencies | **Yes: 78 (7)** |
| | Gender roles have an influence on mental health | **Yes: 89 (8)** |
| | Competencies have an influence on mental health | **Yes: 89 (8)** |
| Hypothesised causal influences starting from gender norms (social environment) | Gender norms have an influence on gender attitudes | **Yes: 89 (8)** |
| | Gender norms have an influence on mental health | **Yes: 89 (8)** |
| | Gender norms have an influence on gender roles | **Yes: 100 (9)** |
| Gender roles | Agreement with operationalisation of gender roles: time spent on gender-typed activities | **Yes: 78 (7)** |
| Social position variables forming an intersectional lens | Relevance of the proposed social positions for adolescents | **Yes: 78 (7)** |
| Section C: Social environment | | |
| Inclusion of actors most likely to influence gender attitudes | Family | **Yes: 100 (9)** |
| | Peers | **Yes: 100 (9)** |
| | School environment | **Yes: 100 (9)** |
| | Influencers | **Yes: 89 (8)** |
| | Social media | **Yes: 100 (9)** |
| Social environment level for only consented actor | Family – **Household** | **100 (9)** |
| | Peers – **Community** | **100 (9)** |
| | Influencers – **Digital** | **100 (8)** |
| | Social media – Community - **Digital** | **89 (8)** |

Bold text = above defined consensus of ≥70% (fairly and very important)

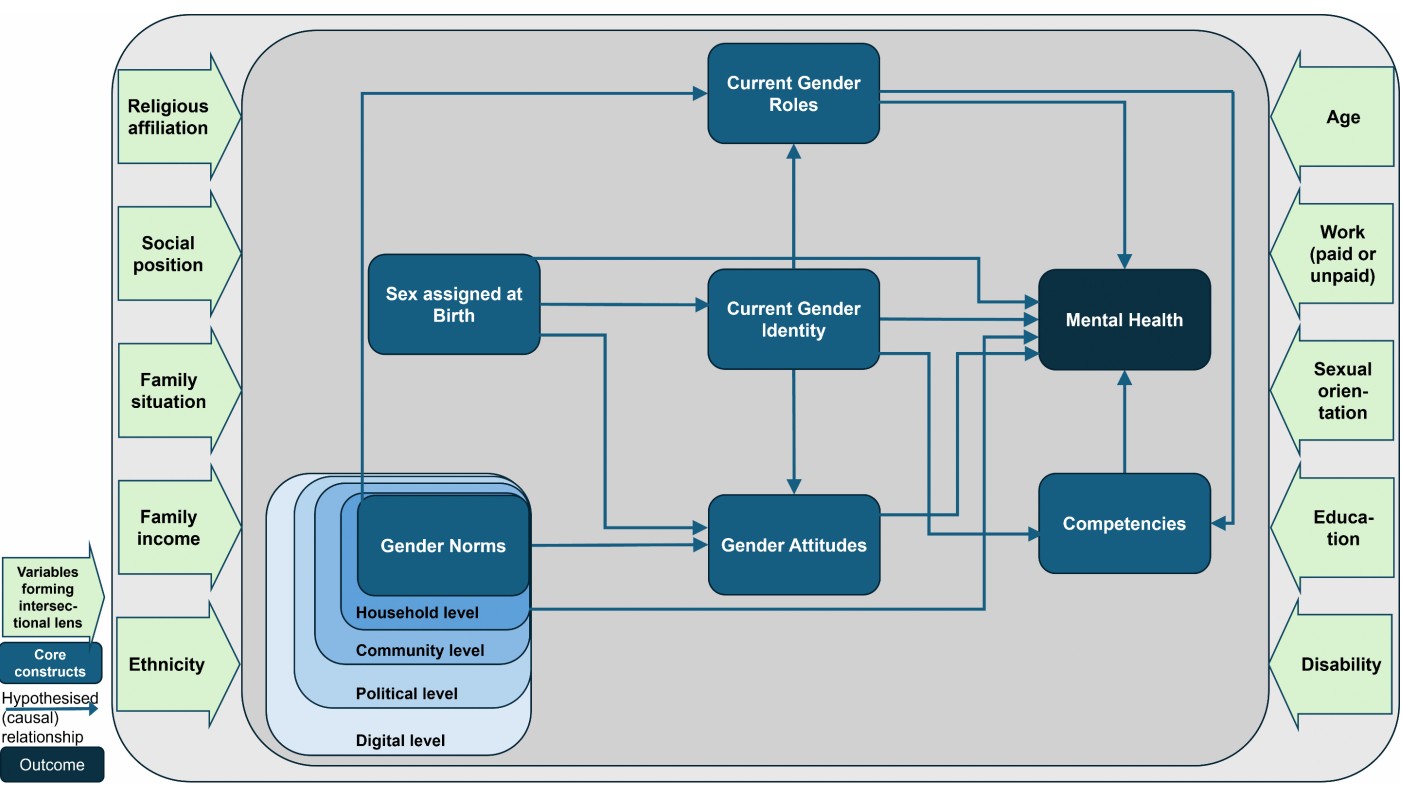

**Fig 2. Consented final conceptual framework developed in three Delphi rounds.**

for adolescents. However, they were not viewed as complete to form an intersectional lens. Moreover, actors such as family, peers, the school environment, media, social media and influencers were selected and assigned by consensus to the different social environment levels (see Fig 2). Despite the inclusion of the political level reaching a consensus in the second round, no actor could be assigned to this level.

The qualitative comments concerning the statements about the included aspects in the framework revealed that the term individual gender norms should be changed to gender attitudes and that there should be a definition for gender roles added. Two participants criticised the operationalisation of gender roles, stating that time spent should be complemented by attitudes toward these roles. However, we could not consider this criticism for this consented version of the conceptual framework, because the operationalisation reached a consensus. Propositions for additional social position variables were migration history instead of ethnicity, family wealth, rural/urban, language spoken in comparison to the dominant language in the region, adverse childhood experiences, country of origin (own or parents), and a clarification of the variable "social position" was asked. The participants further pointed out that the framework should be used as a guide and as a flexible framework where different elements or different perspectives, such as the time-course perspective, can still be added depending on the research questions or the specific context in which the framework is used. In Table 6, we display a final list of the included constructs and gender approaches along with their operationalisations.

The conceptual framework in Fig 2 shows the hypothesised interplay between the six gender-related constructs: gender norms, sex assigned at birth, gender identity, gender roles, gender attitudes, competencies; and the construct of mental health outcomes. The ten variables in light green form the intersectional perspective and take at the same time structural aspects and mechanisms of privilege and disadvantage into account. The variable "social position" refers to different social positions that an individual can occupy in a society, e.g., a caste or a social class. While the items in light

**Table 6. List of included constructs, gender approaches and their operationalisations in the conceptual framework.**

| List of constructs, gender approaches and their operationalisations | |
| --- | --- |
| Constructs | |
| Sex assigned at birth | No conceptual operationalisation |
| Gender identity | No conceptual operationalisation |
| Mental health | Mental, social and physical well-being<br>Connectedness<br>Depressiveness<br>Happiness<br>Body image<br>Resilience<br>Risky behaviour |
| Gender attitudes | *Domains:*<br>Sexual and relationship attitudes<br>Body and appearance attitudes<br>Behaviour attitudes<br>Education attitudes<br>Performance attitudes<br>Mobility attitudes<br>Career attitudes<br>*Type:*<br>Descriptive attitudes<br>Prescriptive attitudes |
| Gender roles | Time spent on gendered activities |
| Competencies | Interpersonal relationship skills<br>Mental health literacy<br>Respect and empathy<br>Agency skills<br>Coping skills |
| Gender norms (actors in social environment levels) | Family (Household level)<br>Peers, School (Community level)<br>Media, Social media, Influencers (Digital level) |
| Gender approaches | |
| Intersectionality approach | Reflected by multiple social position variables: Disability, education, sexual orientation, religious affiliation, age, social position, family situation, family income, ethnicity, work (paid or unpaid) |
| Multilevel approach | Reflected by several social environment levels: household level, community level, political level, digital level |
| Multidimensionality approach | Reflected by developing sex and gender concepts with different dimensions (sex assigned at birth, gender identity, gender roles) |
| Gender power relations lens | Reflected by multiple social position variables (see intersectionality approach) |

blue indicate the operationalisation of the core constructs, they represent different mental health outcomes or proxies for mental health that may be affected by different gender variables.

## Discussion

Based on a Delphi study with experts from different disciplines and cultural backgrounds, the experts in our Delphi panel developed a multilevel and multidimensional conceptual framework with an intersectional and a power relations perspective. They reached consensus on gender norms (of the social environment), gender attitudes, gender roles, gender identity, sex assigned at birth and mental health outcomes. These constructs are central sex/gender constructs to analyse the gendered pathways to adolescent mental health. Moreover, the experts identified a range of competencies that are required for the adolescent to deal with the gender norm expectations of their social environment. The conceptual

framework has a hypothesis-generating nature and is a first step in approaching the gendered pathways of mental health for the age group of adolescents and should therefore be understood as a flexible tool that can be adapted according to the specific sociocultural context and research objectives.

A conceptual framework with the combination of gender roles, gender norms, gender attitudes and competencies in relation to mental health is relatively new. To our knowledge, only two studies exist that cover this research field, namely the GEAS [32] and the GAGE study, with the GAGE study providing the sole conceptual framework for this research area [33,34]. Apart from this, various aspects of the conceptual framework are still understudied in the literature. The conceptual framework of GAGE does not show how the included aspects in their project stand in relation to each other, nor does it give insights into (causal) assumptions. The constructs that are different in our framework compared to the framework from GAGE are the combination of gender roles and gender attitudes, as well as competencies and the focus on mental health. Furthermore, the framework from GAGE does not illustrate sex assigned at birth and gender identity, since its focus is not solely on gender. We would also argue that our framework is more related to theoretical aspects. Since we combined theoretical aspects in a new way, this conceptual framework is the first step to disentangle the specific gendered pathways to mental health.

In the following, we highlight the main areas of consensus, contradiction, and ambivalence. Mental, social and physical well-being (mental health outcome); critical reflection skills (competency); and coping skills (competency) reached the highest consensus with over 90%. Self-efficacy and sense of coherence (mental health outcomes) and navigation (competency) had no clear direction on the importance scale. The item navigation (competency) was also viewed critically by one expert, suggesting that it is not necessarily an ability to navigate in only one social environment but rather to choose and adapt between different social environments. However, the proposed competencies received the most positive feedback in the comments. The items gender expression, decolonial lens, media literacy and self-awareness were the most contrasting with 50% of the panelists wanting to include and 50% wanting to exclude the item. In our Delphi survey, these contrasting views may also result from the fact that we asked for the most relevant constructs due to the quantitative character of the conceptual framework. This means that the views might not be opposing in themselves but rather a consideration of including only the most essential items.

The most debatable parts of the framework might be the gender attitudes categorisation and the gender roles operationalisation. Gender attitudes are subject to change, and are closely connected to culture [17], which is why we were reluctant to propose specific gender attitudes and proposed domains instead. Although this makes our framework less specific, it offers scope for flexible operationalisation. This means that the operationalisation of the constructs can be adapted, such as to the sociocultural context, data availability, or developmental stages of adolescents. Moreover, to date, we do not have a set of gender attitudes that is comparable across cultures [35]. For gender roles, the same applies. There is a high inconsistency and unclarity of terms and operationalisations for gender roles, such as gender role identity, gender role attitudes, gender role beliefs, gender role ideology, gender role stress, gender role socialisation, gender role conflict and gender role self-concept, among others. These are measured by different scales, namely Masculinity and Femininity (included in this scale: BEM Sex role inventory) [36], Traditional Masculinity-Femininity scale [37], Attitudes toward Women Scale [38], Social Roles Questionnaire [39], Gender Role Attitude Scale [40] and the Gender Role Belief Scale [41], among many others. Moreover, the scales Conformity to Masculine Norms Inventory, Conformity to Feminine Norms Inventory, Femininity Ideology Scale, and the Personality Attributes Questionnaire were all mentioned as operationalisations of gender roles [42]. This demonstrates huge gaps and inconsistencies in the literature and the conflation of the terms of gender roles and gender norms. To our knowledge, the only gender role scale that addresses the age group of adolescents is the Children's Sex Role Inventory which was adapted from the BEM Sex Role Inventory [36].

Notably, sexual orientation, which is viewed as a central aspect of gender in gender studies and gender epidemiology [43,44], did not reach a consensus. However, the experts may have seen sexual orientation as an aspect that belongs to the variables that form the intersectionality perspective [43], as is also the case in the conceptual framework of Bolte et al.

(2021) [31]. An important criticism during the Delphi survey was that some of the proposed mental health outcomes are not 'real' mental health outcomes, such as body image etc. but rather determinants or risk factors of mental health. Nonetheless, we decided to keep those items as proxies for mental health outcomes when some 'real' mental health outcomes are unable to capture subtle changes in gender norms. For the competencies of adolescents, it was criticised that it is very subjective what mentally healthy is and therefore also which competencies are necessary for that.

Potential implications for theory and practice following further validation of the framework include an intersectional perspective in designing more tailored interventions, workshops, policies, educational or community-based programs related to adolescents' gender and/or mental health or community gender norms. The conceptual framework could also guide variable selection for secondary data analysis within this research area.

## Strengths and limitations

A significant limitation is that, apart from the inclusion of a few early career professionals in the Delphi study and an expert on adolescence during the development of the first Delphi round, we have not yet incorporated adolescents' perspectives. As this carries the risk that the perspectives or needs of adolescents are not yet fully met, it is an important next step to validate the framework among adolescents using secondary data analysis or to complement and further expand the framework through qualitative interviews with adolescents.

The Delphi technique involves several key methodological decisions and design considerations. However, there is a lack of quality criteria for Delphi studies, especially for different Delphi types. This gap creates uncertainty about how exactly the process of developing a conceptual framework with the Delphi technique should be structured. To adhere to reporting guidelines and enhance the rigour and reliability of our empirical results [19,21], we discuss our methodological considerations in the following.

The *composition of the expert group* is one of the aspects affecting the validity and the quality of the results. In the literature, the concept of the 'wisdom of crowds' has been criticized for being applicable to all groups, rather than being limited to groups of experts [19,22]. We have included experts with research knowledge and people with knowledge of policy or development, services and implementation. We could thus argue that, unlike lay experts, we have involved experts by profession, albeit the experts did not necessarily have expertise in all subtopics and it is difficult to measure experience quantitatively [45]. The subjective rating of the experts indicated, however, that they feel fairly to completely confident on average about the topics under investigation. Furthermore, the results show that we generated a diverse and highly international sample with work experience in different countries, different age groups and gender-diverse experts. Women were overrepresented in our study, possibly indicating that women are more likely to work in areas that tackle discrimination. Furthermore, we employed a continuous sample, i.e., we used the same sample in all three Delphi rounds. Although this may have had an impact on the drop-out rate, we believe that the interdisciplinary topic can be best addressed with the same sample that generates cross-disciplinary knowledge throughout the Delphi rounds.

Quasi-*anonymity* was maintained throughout the whole Delphi process, i.e., the panelists are known to the researcher but remain anonymous to each other. We could not provide full anonymity (panelists also unknown to the researcher) due to the feedback provision via email. However, the introduced pseudonyms contributed to anonymity during the analysis of the results. Only when we fed back the results to the participants did we combine the names and the pseudonyms. The online survey could have also impeded the exchange of ideas and more nuanced views on the framework.

Although controlled feedback is a fundamental aspect of the Delphi methodology, the type of feedback that is provided to the participants widely differs among Delphi studies [46]. In many cases, Delphi studies do not report individual feedback in combination with group feedback [19] and only provide quantitative statistical feedback [19]. This is why we provided the experts with individual and group feedback as well as feedback from the quantitative and qualitative results.

Another critical quality criterion for validity is the *handling of divergent judgements*. We addressed non-consensus of an item as follows: if a questionnaire item did not reach consensus both in the first and second rounds, we excluded it. We

thus assumed that a non-consensus of the same item in two rounds is not essential enough to keep the item. However, we could not repeat newly introduced items in the third round due to the restriction of the Delphi survey to a maximum of three rounds. Furthermore, it is important to note that we did not set a particularly high threshold for consensus (≥70%) because of the exploratory nature of the topic. Those aspects that did not reach this rather moderate threshold were again interpreted as not substantial enough for inclusion in the framework.

Another aspect worth mentioning is that we did not stop the Delphi survey based on achieved consensus, but rather restricted the survey to three Delphi rounds a priori. This was due to the expected time load for experts, a point we can confirm after completing the Delphi study. With three Delphi rounds, we find ourselves in the most common number of rounds in Delphi studies [19]. The restriction to develop the framework within three rounds affected the type of questions we asked in the survey. For example, we changed the type of questions (e.g., 'How important do you find this aspect?' to 'Please choose the most important aspects out of this list'). Furthermore, rather than asking the panelists for suggestions on specific aspects and incorporating them into the next round to reach consensus, we pre-identified aspects that we considered useful as response options. Nevertheless, we paid particular attention to ensuring that the panelists were able to comment on and criticise the proposed aspects. However, we did not consent to the final framework as we still had to integrate the results of the third round. Nonetheless, we asked the panelists in the third round to comment on the framework we presented at that stage.

Lastly, the number of experts and the dropout rate impact the validity of Delphi studies. The number of experts employed in Delphi studies greatly varies, but typically between 10 and 100 experts are involved [30,45,47,48]. The recommended sample size also depends on the issues addressed in the study [47,49]. For example, fewer than ten panelists tend to be used in Delphi studies with a very specific focus or a research objective that requires highly specialised experts [30], such as studies using Delphi for framework development [24,50].

In our study, we are very close to this range, with nine experts who completed all three Delphi rounds. For our study, we found that the complex and theoretical topic may have led experts to be reluctant to participate or that potential panelists may have misinterpreted the study thinking that they needed expertise in more than one topic. Although the Delphi technique does not intend to produce statistically significant and generalisable results [51], the small number of panelists nevertheless raises questions about the validity and reliability of the results. However, the diversity in the demographic characteristics of our panel allows for greater variability in the perspectives of the panelists, ensures that opinions come from multiple independent sources [46,52], and may ultimately achieve a higher generalisation of consensus [45,53]. Throughout the Delphi rounds, we saw a dropout rate of 40% after the first round and another 17% after the second round. We had anticipated such a high dropout rate for our study, because the available time of experts is low while the complexity of the research problem and the heterogeneity of the sample are high.

Furthermore, we cannot exclude a researcher bias, since the construction and phrasing of the questionnaires and the interpretation of the qualitative arguments may have been different for other researchers. To minimize bias, we discussed the development of each questionnaire in the research team [51].

## Conclusion

This study explored the views of experts from different disciplines, backgrounds and geographical locations on the interplay of key constructs in the gendered pathways of health, using the Delphi technique. The panel identified sex assigned at birth, gender identity, competencies, gender roles, gender attitudes, gender norms and mental health as core constructs. They also formed hypotheses about the relationships between these constructs, suggesting how one may influence another. Furthermore, the panel selected four gender approaches as crucial perspectives for the framework. In addition, four social environment levels were included in which actors as carriers of gender norms influence the gender socialisation process of adolescents.

We believe that this framework has a hypothesis-generating nature and can be a starting point for more theory-driven research in this field. Our framework is more nuanced and adaptable than the few other frameworks on this topic and can shed light on more concrete gendered pathways to mental health. It is intended to be adapted to the specific sociocultural context and should be further refined and validated based on sociocultural contexts, new knowledge and different research methods. Secondary data analysis could be particularly useful for validating the conceptual framework and integrating the lived realities of adolescents, but qualitative interviews, particularly with adolescents, could also complement and further expand the framework.

## Supporting information

**S1 File. CREDES recommendations (Guidance on Conducting and Reporting Delphi Studies).**
(DOCX)

**S2 Table. Results of the proposed gender attitudes in the first Delphi round.**
(DOCX)

**S3 Table. Consensus percentage of the quantitative questionnaire items of Delphi round 3 (n = 9).** Full length.
(DOCX)

**S4 Table. Questionnaire Delphi survey round 2.**
(DOCX)

**S5 Table. Questionnaire Delphi survey round 3.**
(DOCX)

## Acknowledgments

We would like to express our gratitude to all the experts who generously shared their time, expertise and insights in the development of the conceptual framework. Their contributions have been instrumental in shaping the foundation of this publication.

## Author contributions

**Conceptualization:** Anita Alaze, John Grosser.

**Data curation:** Anita Alaze.

**Formal analysis:** Anita Alaze, John Grosser.

**Methodology:** Anita Alaze.

**Project administration:** Anita Alaze.

**Supervision:** Céline Miani.

**Validation:** Anita Alaze.

**Visualization:** Anita Alaze.

**Writing – original draft:** Anita Alaze.

**Writing – review & editing:** John Grosser, Oliver Razum, Céline Miani.

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
