## [Decision Letter · Decision Letter 0]

13 Oct 2025

Dear Dr. Alaze,

Thank you for submitting your manuscript to PLOS ONE. After careful consideration, we feel that it has merit but does not fully meet PLOS ONE’s publication criteria as it currently stands. Therefore, we invite you to submit a revised version of the manuscript that addresses the points raised during the review process. Please carefully address requested revisons, particularly regarding methodological and theoretical concerns. 

We look forward to receiving your revised manuscript.

Kind regards,

Marianna Mazza

Academic Editor

PLOS ONE

Journal Requirements:

4. In the online submission form, you indicated that all relevant data are within the manuscript and its Supporting Information files. We have removed personal and sensitive data due to the small sample size and the anonymous nature of the Delphi study. These data are available on reasonable request from the corresponding author.

6. Thank you for stating the following financial disclosure:

This paper is part of a PhD and was thereby funded within the scope of the Junior Research Group GendEpi (Gender Epidemiology) within the Department of Epidemiology & International Public Health, School of Public Health, Bielefeld University.

8. We note that there is identifying data in the Supporting Information file <Ethikantrag Genehmigung_EUB-2023-111-S.pdf, Submission PLOS ONE Paper 2.zip>. Due to the inclusion of these potentially identifying data, we have removed this file from your file inventory. Prior to sharing human research participant data, authors should consult with an ethics committee to ensure data are shared in accordance with participant consent and all applicable local laws.

-Location data

Reviewers' comments:

Reviewer's Responses to Questions

**Comments to the Author**

1. Is the manuscript technically sound, and do the data support the conclusions?

Reviewer #1: Yes

Reviewer #2: Yes

Reviewer #3: Partly

2. Has the statistical analysis been performed appropriately and rigorously?

Reviewer #1: Yes

Reviewer #2: Yes

Reviewer #3: Yes

3. Have the authors made all data underlying the findings in their manuscript fully available?

Reviewer #1: No

Reviewer #2: No

Reviewer #3: Yes

4. Is the manuscript presented in an intelligible fashion and written in standard English?

Reviewer #1: Yes

Reviewer #2: Yes

Reviewer #3: Yes

Reviewer #1: The research paper titled "Gender and Mental Health of Adolescents: A Conceptual Framework Developed in a Delphi Study" by Anita Alaze and colleagues explores the intersection of gender norms and mental health during adolescence. It aims to address the lack of a theoretical framework that conceptualizes the relationship between gender norms, attitudes, and mental health outcomes.

Key Highlights:

Objective: The study develops a conceptual framework to illustrate the interplay of key constructs in the gendered pathways of adolescent mental health using the Delphi technique2.

Methodology:

An international panel of 21 experts participated in three rounds of surveys.

The study identified core constructs, operationalized them, and hypothesized their relationships.

A 70% consensus threshold was used.

Findings:

Core constructs include gender norms, sex assigned at birth, gender attitudes, roles, competencies, and identity.

The framework incorporates intersectional, multidimensional, and multilevel approaches, along with a power relations lens.

Four social environment levels—household, community, political, and digital—were identified2.

Conclusion: The framework links six key constructs, four gender approaches, and four social environment levels to adolescent mental health. Future research is encouraged to validate and apply these relationships. This paper provides a valuable foundation for understanding the gendered pathways to mental health in adolescence.

The practical implications of this study's findings can significantly influence multiple domains, especially in shaping policies, programs, and interventions aimed at improving adolescent mental health. Here are some key applications:

Mental Health Programs:

The framework provides a structure for designing mental health interventions tailored to address the gendered experiences of adolescents.

By recognizing the impact of gender norms and roles, practitioners can develop targeted therapies or workshops for adolescents struggling with gender-related mental health challenges.

Education Policies:

Schools can integrate the conceptual framework into their curricula to promote gender-inclusive environments.

Educators can create awareness among students about how gender attitudes and norms affect mental health.

Community Engagement:

Community-based programs can use this framework to advocate for cultural changes and challenge harmful gender norms impacting adolescents' mental health.

Digital Platforms:

The study highlights the influence of digital environments, suggesting opportunities to use online tools for mental health education and support.

Social media campaigns can be designed to promote healthier gender attitudes and provide resources for mental health care.

Policy Development:

Policymakers can apply the framework when drafting legislation focused on adolescent mental health, ensuring gender is a central consideration.

This could include funding for gender-sensitive mental health services and training for practitioners.

Intersectional Approaches:

The findings encourage the use of intersectionality in addressing mental health, helping practitioners and researchers to cater to diverse gender identities and socio-economic contexts.

By implementing these strategies, the conceptual framework developed by Anita Alaze's study offers transformative opportunities to advance adolescent well-being and mental health equity.

Applying the findings from Anita Alaze's study may present several challenges due to the complexity of factors involved. Here are some key potential obstacles:

Cultural Resistance:

Deeply ingrained gender norms may face resistance from communities or institutions reluctant to change. Advocacy efforts may be slow to gain acceptance in certain cultural contexts.

Implementation Across Diverse Contexts:

Adolescents' experiences with gender and mental health vary widely based on culture, socio-economic background, and personal identity. Adapting the framework to meet these diverse needs could be complex.

Interdisciplinary Coordination:

Effectively addressing gendered pathways in mental health requires collaboration between educators, policymakers, healthcare providers, and community leaders. Aligning these sectors may be logistically challenging.

Resource Limitations:

Implementing gender-sensitive programs and interventions demands funding, trained professionals, and infrastructure, which may not be readily available in all regions.

Intersectionality Challenges:

The framework emphasizes intersectionality, but addressing multiple overlapping identities (e.g., gender, socioeconomic status, ethnicity) requires nuanced approaches that may not be well-established yet.

Digital Risks:

While the digital space offers opportunities for outreach, it also poses risks like cyberbullying, misinformation, and exposure to harmful content, which can undermine the benefits of mental health interventions.

Measurement and Validation:

The framework needs empirical testing and validation in real-world settings, which can be time-consuming and methodologically challenging.

Stigma and Awareness:

Persistent stigma around mental health and gender issues may deter adolescents from seeking help or participating in programs.

Overcoming these challenges will require concerted efforts to tailor interventions, promote cultural sensitivity, and secure necessary resources. It’s a reminder that progress in areas like mental health and gender equity is gradual but profoundly impactful.

The research paper concludes with the presentation of a conceptual framework that highlights the intricate relationship between gender norms, attitudes, and adolescent mental health. By identifying core constructs and employing a multidimensional, intersectional approach, the study provides a foundational tool for understanding the gendered pathways to mental health outcomes.

Key Takeaways from the Conclusion:

Framework Validation: The study underscores the importance of empirically testing and validating the proposed framework in diverse contexts to enhance its applicability.

Call for Action: It advocates for researchers, policymakers, educators, and mental health practitioners to incorporate gender-sensitive approaches in their work.

Future Directions: It encourages further exploration and refinement of the identified constructs and relationships to better support adolescent mental health globally.

Impactful Change: The framework sets the stage for challenging harmful gender norms and creating environments that foster equity and mental well-being.

The conclusion of this project highlights both its theoretical contributions and its potential for real-world impact, paving the way for meaningful interventions in adolescent mental health.

Reviewer #2: Dear authors,

Your contribution to knowledge is commendable. However, there is need to clearly raise a research question to guide your inquiry.

Although, it was later identified that your sample size is 20 expert. It will be fine if you can have a separate subheading for population and sample for clarity.

Lastly, there is need for you to mention the implications of your findings for theory and practice.

Reviewer #3: While the objective of producing a theoretical framework for an understudied area is clear, several methodological and substantive concerns limit the paper's current contribution and warrant careful reconsideration. The framework organizes constructs across individual, relational, community, and societal levels, highlighting intersectionality and the complexity of gendered pathways - this is a strength of the paper, as is the synthesis attempt (to consolidate fragmented literature and expert opinion into one framework provides a starting point for discussion and hypothesis generation). However, there are methodological concerns. First, there is a mismatch between method and subject matter. For adolescent gender norms (including links to mental health), a substantial empirical literature exists - including large-scale survey and qualitative studies (e.g., GEAS (which the authors mention), GEM scale, several publications in the Lancet of global gender norms analyses) - which could and should have informed framework development. By relying almost exclusively on expert opinion - and without adolescent participation - the study risks producing constructs that are unanchored from lived experiences. Second, simplifying context-specific insights into standardized constructs. This is particularly problematic given evidence that gender norms vary across cultural contexts, age groups, and social environments. The authors do reference GEAS and GAGE as existing work but does not engage with their key findings. For example, GEAS demonstrates that adolescents' lived experiences reveal site-specific variations in gender norms, with age and gender-specific differences and culturally contingent social hierarchies. This contradicts the assumption that standardized, expert-defined constructs can fully capture gendered experiences. GAGE provides a conceptual framework that does not capture sex assigned at birth or gender identity (that the authors acknowledge), and its constructs are empirically grounded rather than purely theoretical. The authors claim that their framework is "more related to theoretical aspects" which ignores the importance of empirical validation and the context-specific variations documented in these studies (as well as others). By not fully incorporating these empirical insights, the Delphi framework risks representing expert assumption rather than adolescent realities. The methodological references in the paper (e.g., Niederberger & Spranger, Jorm etc) outline proper Delphi procedures but simultaneously highlight that Delphi provides one of the lowest levels of evidence and should only be applied where higher-quality empirical methods are infeasible. Given the extensive literature on adolescent gender norms, the choice of Delphi is methodologically questionable and should be further explained. The framework's utility for guiding interventions or measurement development depends entirely on future validation. However, empirical studies suggest that starting with adolescent voices would likely yield substantially different constructs, reducing the framework's preliminary credibility. My suggested improvements are: 1) Engage existing empirical literature, beyond GEAS and GAGE, and include key studies such as Weber et al, 2019 (Lancet), Kagesten et al, 2016 (mixed-methods review), and various GEM scale applications provide quantitative and qualitative evidence of age, gender, and context-specific variations that could have informed construct selection. 2) Ground constructs in adolescent experience: incorporating youth perspectives either through mixed-methods consultation or secondary data analysis would enhance the conceptual validity of the framework (and/or include the reason as to why this was not able to be done/could not be done, rather than it being just a "limitation"). 3) Clarify theoretical vs. empirical positioning: Explicitly framing the framework as a hypothesis-generating exercise rather than an authoritative depiction of gendered pathways would set accurate expectations for subsequent research. 4) Contextual sensitivity: The framework should highlight that constructs may manifest differently across cultures and developmental stages (i.e., clarify why/how this would be flexible and adaptable as mentioned in the paper). Overall, the theoretical ambition is clear, and the Delphi method is correctly applied procedurally, but the framework is fundamentally limited by its lack of engagement with empirical evidence and adolescent perspectives. I recommend major revision, with particular attention to integrating existing empirical studies, explicitly addressing context specificity, and clarifying the framework's provisional, hypothesis-generating nature. This paper can provide great value as a starting point for theoretical discussion, but it must be revised to incorporate empirical grounding and adolescent perspectives. Without this, the framework risks codifying expert assumptions rather than capturing the lived realities of adolescence, and what the evidence base already says, which limits its relevance for research and intervention in the field of gender and mental health.

**Do you want your identity to be public for this peer review?** For information about this choice, including consent withdrawal, please see our Privacy Policy

Reviewer #1: **Yes: ** YOGINI DOLKE

Reviewer #2: **Yes: ** Dr. Sulaimon Adewale

Reviewer #3: No

---

## [Author Response · Author response to Decision Letter 1]

2 Nov 2025

Thank you very much for your comments. We have provided our point-by-point comments in the uploaded rebuttal letter. Thank you!

---

## [Decision Letter · Decision Letter 1]

24 Nov 2025

Gender and mental health of adolescents: a conceptual framework developed in a Delphi study

PONE-D-25-01695R1

Dear Dr. Alaze,

We’re pleased to inform you that your manuscript has been judged scientifically suitable for publication and will be formally accepted for publication once it meets all outstanding technical requirements.

Kind regards,

Marianna Mazza

Academic Editor

PLOS ONE

Additional Editor Comments (optional):

Reviewers' comments:

Reviewer's Responses to Questions

**Comments to the Author**

Reviewer #3: All comments have been addressed

2. Is the manuscript technically sound, and do the data support the conclusions?

Reviewer #3: Partly

3. Has the statistical analysis been performed appropriately and rigorously?

Reviewer #3: N/A

4. Have the authors made all data underlying the findings in their manuscript fully available?

Reviewer #3: Yes

5. Is the manuscript presented in an intelligible fashion and written in standard English?

Reviewer #3: Yes

Reviewer #3: (No Response)

**Do you want your identity to be public for this peer review?** For information about this choice, including consent withdrawal, please see our Privacy Policy

Reviewer #3: No

---

## [Editor Report · Acceptance letter]

PONE-D-25-01695R1

PLOS One

Dear Dr. Alaze,

I'm pleased to inform you that your manuscript has been deemed suitable for publication in PLOS One. Congratulations! Your manuscript is now being handed over to our production team.

Kind regards,

on behalf of

Dr. Marianna Mazza

Academic Editor

PLOS One